# Optimal Output Feedback $H_{\infty}$ Torque Control of a Wind Turbine Rotor using a Parametrically Scheduled Model

Dana Martin<sup>1</sup>, Kathryn Johnson<sup>1</sup>, Christopher Bay<sup>1,2</sup>, Daniel Zalkind<sup>2</sup>, Lucy Pao<sup>2</sup>, Meghan Kaminski<sup>3</sup>, and Eric Loth<sup>3</sup>

<sup>1</sup>Colorado School of Mines Golden, CO 80401
 <sup>2</sup>University of Colorado Boulder, CO 80309
 <sup>3</sup>University of Virginia, VA 22904

Correspondence to: Dana Martin (dmartin@mymail.mines.edu)

Abstract. Wind turbines are nonlinear, time-varying systems that are subject and sensitive to model parameter variations and a stochastic wind field. For such applications, Linear Parameter Varying (LPV) control provides a state-space approach to designing nonlinear controllers with robust performance. LPV uses multi-input multi-output (MIMO) model with a guaranteed limit on the exogenous disturbance's gain with respect to performance signals. A robust matrix inequality synthesis of an  $H_{\infty}$ 

5 based performance LPV controller using a parametrically varying model will be developed with the goal of obtaining a torque controller with drive-train damping properties. The technique guarantees a-priori performance values and closed-loop stability for the simplified model, and provides a systematic tuning procedure to adjust controller performance.

# 1 Introduction

- Wind turbine technology has emerged as the most competitive form of renewable energy production (REN21, 2017, 2016)
  during the green push over the past two decades, and the evolution of wind energy control systems continue to enable the advancement of turbine technology. Wind energy increased its penetration into the electricity grid around the world with a record 63 GW of wind generation added during 2015 and 55 GW during 2016, equating to a 22% and 12% respective growth on the world market compared to previous years' statistics (REN21, 2017, 2016). Wind offers a renewable alternative to carbon-based energy sources with the promise of lowering carbon emissions by 2,700 tons of CO<sub>2</sub> per year per 1.5 MW
- 15 turbine, which is the equivalent of planting 4 square kilometers of forest each year (REN21, 2016). While renewable growth had record numbers during 2015 with 2016 coming in a close second, the Energy Information Administration (EIA) predicts that the U.S. wind power capacity would have to reach more than 300 GW to achieve the 20% wind-generated electricity by 2030, which would result in a displacement of 50% electric utility natural gas consumption and 18% coal consumption (of Energy Efficiency and Energy, 2008). To meet 20% by 2030, the U.S. needs to increase its wind power penetration by
- 20 265% of its current capacity, equivalent to an additional 218 GW of wind power (REN21, 2016, 2017)(of Energy Efficiency and Energy, 2008). With so much growth still to come in the next decade, the market is ripe for emerging technologies to further improve the viability and impact of wind energy on the world market by further decreasing the levelized cost of energy

(LCOE). One such technology with the ability to improve existing wind turbine operation, in addition to enabling advanced turbine technology is the control system.

Nonlinear dynamics and modeling uncertainties of wind turbines make non-linear and gain scheduling methodology necessary for optimal operation across all wind speeds. However, non-linear control architectures do not easily lend themselves to

5 multi-input multi-output (MIMO) controller synthesis, and lack the theory for a-priori guarantees of stability (Shamma, 1988). To address the immediate industry needs of improving performance prediction and decreasing the number of under-performing, short-lived turbines (Fields et al., 2016), Linear Parameter Varying (LPV) control can be utilized for both increasing performance and the longevity of turbine life (Adegas and Stoustrup, 2012; Inthamoussou et al., 2016; Sloth et al., 2011).

When formulated using a state-space representation, it is conducive to MIMO control synthesis satisfying multiple control

- objectives and robust performance. This formulation allows the designer to ensure stability given plant parameter uncertainty, and uncertainties in the measured scheduling parameter (Shamma, 1988; Zhao and Nagamune, 2017; Sato and Peaucelle, 2013; Sato et al., 2010). LPV control theory can utilize  $H_2$  and  $H_{\infty}$  (Levine, 2011) concepts during the construction of the optimization problem, giving it the ideal combination of the robustness of H control and optimal performance across a wide operating envelope. However, the range of the operating envelope also contributes to the nonlinearity of the optimization
- model, and with below-rated operation of a variable speed wind turbine, solving the optimization is not always trivial due to the dimensionality of the problem. For this reason, methods such as the gridding technique and slack variable approach can be used to create feasible solutions to once infeasible optimization problems allowing existing conic solver algorithms (Sturm, 1999) to find globally feasible solutions (Ostergaard et al., 2009).

Most recent work regarding LPV control of wind turbines has focused on variable pitch systems due to their prevalence in the industry and increased controllability. A detailed derivation of LPV control schemes for Region 2 (below rated) and Region

- the industry and increased controllability. A detailed derivation of LPV control schemes for Region 2 (below rated) and Region 3 (above rated) wind conditions is given in (Bianchi et al., 2006). The authors develop a LPV model that is scheduled on the wind speed, generator speed, and blade pitch angle. This leads to the development of conventional single-input single-output (SISO) controllers for turbine speed control in Region 3 and generator torque control in Region 2. Bianchi, et al. then develop a MIMO controller that gives increased power performance (smoothed generator output) without increased pitch actuation while

substantially reducing torque fluctuations. Finally, the authors present a method for combining the two MIMO LPV controllers to provide control over the entire operating envelope.

Others have also proposed LPV controllers for operation across the wind envelope for turbines. A switching LPV controller for a variable speed and pitch turbine was presented in (Lescher et al., 2006). The main objective of this controller was to reduce drivetrain torsion, blade and tower flexion. This control scheme is then compared against conventional PI controllers and a gain scheduled LQG controller, showing reduced mechanical fatigue in the system while maintaining power output.

Further extending the objectives in operating wind turbines, work has been done with LPV-based active power control (APC). In (Inthamoussou et al., 2016), the authors develop an LPV controller within the standard decoupled pitch and torque control structure. The controller is scheduled on pitch and power demand, thus not depending on wind speed measurements. The controller was simulated in FAST on the NREL 5MW benchmark turbine (Jonkman et al., 2009), compared against gain

scheduled PI and  $H_{\infty}$  controllers, the LPV controller outperformed the others in terms of APC while providing a decrease in

aerodynamic loads. While the above works showcase several of the benefits of LPV control for wind turbines, the simulations were done with a traditional three-bladed, upwind turbine. The two-bladed, downwind configurations are unexplored with regard to LPV control.

The contributions of this paper lie in a drive train damper utilizing only output feedback of the generator speed angular velocity as opposed to state-feedback (Darrow et al., 2011), comparisons of controller derivation and performance to existing 5 methods (Ostergaard et al., 2009), detailed modeling, and implementation of a robust linear matrix inequality (LMI) approach to gain scheduled, output feedback torque control of a Segmented Ultra-light Morphing Rotor (Loth et al., 2017). The control synthesis methodology to be developed in the following paper lends itself to a generalized plant that can be modeled using aerodynamic basis functions (Mohammadpour and Scherer, 2012) and simplified linear models to arrive at a working torque controller for a full degree of freedom (DOF), non-linear plant. 10

In this paper the authors will propose, synthesize, and implement an LMI based LPV controller using  $H_{\infty}$  performance specifications for below-rated operation with the aim of increasing power capture at low wind speeds and reducing component loads. The paper will be organized as follows. Section 2 will derive a simplified, two mass model of the rotor and drive train, which will be used in Section 3 to synthesize an optimal LPV controller using an  $H_{\infty}$  performance metric. Section 4 will

15 provide below rated, turbulent inflow simulation results and data analysis to quantify controller performance. Finally, Section 5 gives closing remarks.

#### Modeling 2

#### 2.1 Three state drive-train model

20

In order to implement a model-based controller to damp drive-train oscillations, a simplified turbine model needs to be derived. Using a flexible drive-train model of the low-speed shaft (LSS), gear box with ratio  $N_g$ , and high speed shaft (HSS) as depicted in Figure 1, torque balances on either side of the gear box are performed to develop the relationship between generated aerodynamic torque  $Q_a$ , drive-train torsion and applied generator torque  $\tau_q$ .

Figure 1. The flexible, two mass drive-train model. The section labeled HSS depicts the high speed shaft side of the gear box with a rotational inertia  $I_g$ , angular velocity  $\Omega_g$ , an azimuthal position of  $\theta_g$ . The gear box ratio ( $N_g$ ) section is the gearing connecting the HSS and LSS masses of the drive-train. The section labeled LSS depicts the low speed shaft side of the drive-train connected directly to the rotor with a rotational inertia  $I_r$ , angular velocity  $\Omega_r$  and azimuthal position of  $\theta_r$ . The model assumes a drive-train with no frictional damping.

Figure 1 depicts a two mass model of the rotor, with the gear box connecting the LSS mass (denoted using subscript rot) and the HSS mass (denoted using subscript g). Equation (1) describes the torque balance on the LSS side of the gear box with  $Q_a$  being the aerodynamic torque generated by the wind on the rotor, and (2) describes the torque balance on the HSS side of the gear box where  $Q_{LSS}$  represents the torque applied by the LSS through the gear box.

$$5 \quad I_{rot}\dot{\Omega}_r(t) = Q_a(t) - Q_{LSS} \tag{1}$$

$$I_g \Omega_g = Q_{HSS} - \tau_g \tag{2}$$

The LSS torque can also be computed using the drive-train torsional spring constant,  $k_d$ , torsional damping constant,  $C_d$ , and a difference in azimuthal position ( $\phi$ ) and angular velocities ( $\Omega$ ) of the two masses across the gear box as shown in (3).

$$Q_{LSS} = \frac{k_d}{N_g} (\phi_r - \phi_g) + C_d \left(\frac{\Omega_r}{N_g} - \frac{\Omega_g}{N_g^2}\right) \tag{3}$$

10 The generated aerodynamic torque given an operating point  $(|_{OP})$  can be linearly approximated as a function of rotor speed  $(\Omega_r)$ , blade pitch  $(\beta)$ , and free stream wind speed  $(V_{\infty})$  as

$$Q_a(\Omega_r,\beta,V_\infty) = \bar{Q}_a\Big|_{OP} + \left.\frac{\partial Q_a}{\partial \Omega_r}\right|_{OP} \delta\Omega_r + \left.\frac{\partial Q_a}{\partial \beta}\right|_{OP} \delta\beta + \left.\frac{\partial Q_a}{\partial V_\infty}\right|_{OP} \delta V_\infty. \tag{4}$$

Equation (4) includes the nominal aerodynamic torque,  $\bar{Q}_a|_{OP}$ , for a given operating point plus the sensitivities around that operating point, which vary according to the selected local operating point. The operator  $\delta$  indicates deviation of a variable from its local operating point.

The final form of the 3 DOF system's dynamics formulated into state-space format are given by (5).

$$\begin{bmatrix} \dot{x}_1\\ \dot{x}_2\\ \dot{x}_3 \end{bmatrix} = \begin{bmatrix} \frac{\frac{\partial Q_a}{\partial \Omega_r}|_{OP} - C_d}{I_{rot}} & \frac{-k_d}{I_{rot}} & \frac{C_d}{I_{rot}N_g} \\ 1 & 0 & \frac{-1}{N_g} \\ \frac{C_d}{I_gN_g} & \frac{k_d}{I_gN_g} & \frac{-C_d}{I_gN_g^2} \end{bmatrix} \begin{bmatrix} x_1\\ x_2\\ x_3 \end{bmatrix} + \begin{bmatrix} 0 & \frac{\partial Q_a}{\partial \beta}|_{OP} \frac{1}{I_{rot}} \\ 0 & 0 \\ \frac{-1}{I_g} & 0 \end{bmatrix} \begin{bmatrix} \delta \tau_g \\ \delta \beta \end{bmatrix} + \begin{bmatrix} \frac{\partial Q_a}{\partial V_{\infty}}|_{OP} \frac{1}{I_{rot}} \\ 0 \\ 0 \end{bmatrix} \delta V_{\infty}$$
(5)

where the state vector is given by (6).

$$\begin{bmatrix} x_1 \\ x_2 \\ x_3 \end{bmatrix} = \begin{bmatrix} \delta\Omega_r \\ (\phi_r - \phi_g) \\ \delta\Omega_g \end{bmatrix}$$
(6)

20

15

The partial derivatives of aerodynamic torque are not constant throughout the entire operating range of a wind turbine. Thus, the system dynamics defined in (5) vary according to wind speed, which will be developed in Section 2.2.

## 2.2 Basis Functions

Torque control of a wind turbine is not stationary because the dynamics of the turbine differ according to the inflow velocity. The controller must be able to follow an operating trajectory as opposed to maintaining a single operating point. Since the

partial derivatives of aerodynamic torque (4), with respect to the three independent variables  $(\Omega_r, \beta, V_{\infty})$  vary throughout the operating envelope, they can provide the parametric variation required for LPV controller synthesis. Figure 2 shows these variations as functions of free stream wind speed at selected turbine operating points along the partial load operating trajectory in below-rated conditions.

**Table 1.** Operating points of the turbine used to calculate the operating points and A, B,  $B_w$  dependence on wind speed corresponding to the below-rated trajectory of the turbine.

| $V_{\infty}$ (m/s) | $\Omega_r$ (rpm) | $\tau_g$ (N-m) | $\beta$ (deg) |
|--------------------|------------------|----------------|---------------|
| 2                  | 7.273            | 58             | 0.5           |
| 3                  | 11.72            | 140            | 0.5           |
| 4                  | 15.77            | 274.2          | 0.5           |
| 5                  | 21.97            | 410            | 0.5           |

**Figure 2.** Partial derivative for aerodynamic torque approximation (4). Subplot (a) shows variation of  $\frac{\partial Q_a}{\partial \Omega_r}$  with  $V_{\infty}$ . Subplot (b) shows variation of  $\frac{\partial Q_a}{\partial V_{\infty}}$  with  $V_{\infty}$ . Subplot (c) shows variation of  $\frac{\partial Q_a}{\partial \beta}$  with  $V_{\infty}$ . The portion of the sensitivities corresponding to below-rated operating points is indicated by the shaded gray box.

5 This paper is focused on partial load operation (region II) (Johnson et al., 2006), therefore, only the portion of the sensitivities falling between the cut-in (2 m/s) and rated (5 m/s) wind speeds will be used in the construction of the parameter varying functions. The turbine under study is a scaled design that has resulted in very low wind speed operation. See Section 4.1 for more information. Due to the required assumptions of affine dependence for system scheduling (Ostergaard et al., 2009), only