# Peer review of "Optimal Output Feedback $H_{\infty}$ Torque Control of a Wind Turbine Rotor using a Parametrically Scheduled Model"

_Wind Energy Science, 2018_

## Referee Comment (RC1) · Anonymous Referee #1 · 22 May 2018

Partial review of manuscript "Optimal Output Feedback H_inf Torque Control of a Wind Turbine Rotor using a Parametrically Scheduled Model"

General comments: - Review your overall writing/structuring. Should be improved. - MIMO stands for Multiple-Input Multiple-Output, use capital characters - Subscripts not being an index, should be non-italic. - Search for uses of "will be", "can be". Replace with present tense ("The paper will be organized..." -> "The paper is organized") - Some figures are not nice and/or in high-resolution

0. Abstract Revise complete abstract. Clearly, state the problem, your approach to solve the problem. How you did implement and what the most important results are.

[Figure]

1. Introduction: - Too long first paragraph, shorten this, get to the point more quickly. - When formulated ... robust performance. I don't get this sentence, revise. - One such technology ... the control system. I don't get this sentence, revise. - LPV control theory ... wide operating envelope. What do you mean, I dont understand this. Are H_inf and H_2 techniques used at the same time? - For this reason ... globally feasible solutions. I don't get this sentence, revise. - Call variable pitch systems -> Variable-Speed Variable-Pitch (VSVP) wind turbines. - The controller was ... aerodynamic loads. But the reduced aerodynamic loads were not taken into account during control design, right? What is the cause and consequense of the reduced aerodynamic loads? - The two-bladed, downwind... Now you suggest that you will perform your research on two-bladed turbines, but this does not become clear in the rest of the introduction. - The contributions of this paper... Sentence is unreadable, rewrite. - The following paper -> this paper - Are the aerodynamic basis functions + linearizations obtained and/or based on the full DOF non-linear plant?

2. Modeling - Introduce this section. What will you be doing in what section? - Which wind turbine (model) do you use in this paper? - Why do you use different symbols ($Q\_a$ and $\tau\_g$) for torques? Also $Q\_{xx}$ and tau are not indicated in Figure 1. - Maybe introduce a new symbol for $\phi\_r$-$\phi\_g$ - How do you obtain the aerodynamic torque gradients? From data of a real wind turbine? - Definition of $Q\_{HSS}$ is missing - Directly substitute Eq. (6) in (5). - According to my derivation, the state-space system in (5) is incorrect. For example, according to Eq. (3), you miss a $1/N_g$ for the $C\_d$ term in the (1,1) element. Or (3) is incorrect.

- How do you obtain the partial derivatives? What is your approach? - Why do you only consider 2 to 5 m/s wind? This is just the lower-region of the below-rated region: some wind turbines do not even operate at these wind speeds!

3. Performance vector design - "For this application ... measured scheduling parameter", you stated in you introduction that you would only use output feedback of the generator speed! Also the assumption of 0 error is not really feasible. - Cite Bossanyi

for the drivetrain damper bandpass-filter part. "The Design of Closed Loop Controllers" - Remove Eq. 13. - Figure 4: I don't see a low-pass filter behavior in z(2). - The design of z ... 3 state system. What do you mean here? - Comment on the results in Figure 5, what can be seen? - Complete paragraphm "The loop shaping ... and step response", should be revised. I don't understand what's going on here.

————————————————-

I aborted the review, as a lot of things need to be fixed first. Considering the authors listed, I would expect a higher quality than the submitted manuscript.

---

## Referee Comment (RC2) · Anonymous Referee #2 · 23 May 2018

This paper presents an LPV controller design for a novel rotor. The controller design procedure is explained in detail and seems to be a straightforward implementation of an existing synthesis framework. The LPV modelling step follows similar steps as the work in Bianchi. It is really hard to find the novelty of this work. The paper is rather cumbersome to read and some strange (incorrect?) statements are made:

- Abstract: "For such applications, Linear Parameter Varying (LPV) control provides a state-space approach to designing nonlinear controllers with robust performance" you also synthesis an LPV controller. Why is this a nonlinear controller?

- Abstract: "LPV uses multi-input multi-output (MIMO) model with a.." please check

grammar (e.g Linear Parameter Varying uses . . . what does it mean?)

- Section 2, "only the portion of the sensitivities falling between the cut-in (2 m/s) and rated (5 m/s) wind speeds will be used in the construction of the parameter varying functions" a rated speed of 5 m/s sounds strange. Is that for the novel rotor? If yes, this rotor model should be introduced.

- Section 3, pg 8, line 10, "sensitivity function ($S = (I + Gp)^{-1}$) where Gp is the transfer function from the horizontal inflow wind to rotor angular velocity without any closed-loop control,.." Really strange! How can a wind speed measurement be used in a feedback loop. I really don't understand what is going on here.

- Eq 14 with the previous S, really doesn't make sense. I stopped reading the paper after this point. While the ACC paper from the same authors was easy to read, correct and well-organised this paper seems to be the opposite.

Other minor issues

Pg 2, line 13, "robustness of H control" what do you mean? P2 2, line 21, "a LPV" should be "an LPV" P3, quality of the figure should be improved P8, figure 4 not clear. For performance channels it doesn't make sense to present the phase. All the figures should be checked (readability)

---

## Author Comment (AC1) · 11 Aug 2018

Response to referee 1 comments Thank you to the reviewers and editors for the comments and the opportunity to revise and improve our paper. We have made substantial revisions to the paper to address the comments from both reviewers. In this document, we explain how we addressed specific comments in our revision.

General comments: The resolution of all figures has been addressed. Revise complete abstract: The abstract has been revised as follows: "Wind turbine fatigue damage can be greatly reduced through the application of Linear Parameter Varying (LPV) control architectures as compared to traditional control methodologies. LPV control theory facilitates optimal controller synthesis considering various objectives; in this work, we apply LPV control theory to a MIMO LPV model of a nonlinear turbine plant using Linear Matrix Inequalities (LMIs) and convex optimization solvers to produce the LPV controller. The application of the torque controller to a down-wind, two bladed, scaled, Segmented Ultra-light Morphing Rotor (SUMR) results in improved turbine performance and reduced damage equivalent load (DEL) accumulation during turbulent inflow. The reduction in DEL is attributed to the limit placed on the exogenous disturbances' effect on performance channels stemming from $H_\infty$ LMI formulation."

Introduction: Too long first paragraph, shorten this, get to the point more quickly. Paragraph has been edited to "Wind offers a renewable alternative to carbon-based energy sources with the promise of lowering carbon emissions by 2,700 tons of $CO2$ per year per 1.5 MW turbine, which is the equivalent of planting 4 square kilometers of forest each year (REN21, 2016). While renewable energy growth had record numbers during 2015 the Energy Information Administration (EIA) predicts that the U.S. wind power capacity would have to reach more than 300 GW to achieve the goal 20% wind-generated electricity by 2030, which would result in a displacement of 50% electric utility natural gas consumption and 18% coal consumption (DOE, 2008). To meet 20% by 2030, the U.S. needs to increase its wind power penetration by 265% of its current capacity, equivalent to an additional 218 GW of wind power (REN21, 2016, 2017; DOE, 2008). With so much growth still to come in the next decade, the market is ripe for emerging technologies to further improve the viability and impact of wind energy by further decreasing the levelized cost of energy (LCOE). Implementing advanced control architectures on existing turbines can improve turbine performance without the added capital cost of redesigning and installing upgraded turbine components (Fields et al., 2016)." (pg. 1) One such technology ... the control system. I don't get this sentence, revise. The sentence contained within paragraph 1 has been addressed as given in response 1i. When formulated ... robust performance. I don't get this sentence, revise. Sentence has been revised as "LPV control theory provides guarantees of stability despite the presence of plant parameter uncertainty and/or uncertainties in the measured

scheduling parameter (Shamma, 1988; Zhao and Nagamune, 2017; Sato and Peaucelle, 2013; Sato et al., 2010). It utilizes robust control theory (Levine, 2011) during the construction of the LMI constraints for the optimization problem, providing guarantees of stability in the presence of deviations from model parameters used during control synthesis, and similar guarantees of stability in the presence of exogenous disturbances the plant may be exposed to during operation." (pg. 2) LPV control theory ... wide operating envelope. What do you mean, I don't understand this. Are $H_\infty$ and $H_2$ techniques used at the same time? Sentence has been revised as presented in response 3i. For this reason ... globally feasible solutions. I don't get this sentence, revise. This statement is referring to the complexity of finding feasible solutions in a complex terrain for an optimization problem highly sensitive to parameters such as grid spacing, system mapping, convergence criterion, and performance metric constraints. Sentence has been revised as "To deal with the complexity of the LPV optimization process, the gridding (Wang and Seiler, 2018) technique is utilized, allowing existing conic solvers (Sturm, 1999) to find globally optimal solutions within a highly nonlinear domain and ill-conditioned problem (Ostergaard et al., 2009), in addition to creating finite dimensional LMI's from an originally infinite dimensional LPV model." (pg. 2) Call variable pitch systems -> Variable- Speed Variable-Pitch (VSVP) wind turbines. This suggestion has been incorporated into paper. The controller was ... aerodynamic loads. But the reduced aerodynamic loads were not taken into account during control design, right? What is the cause and consequence of the reduced aerodynamic loads? Aerodynamic loads such as blade root bending moments (flap and edge) were not taken into account during the control design process, correct. On the other hand, the drivetrain torsion was taken into account (the cause) during the controller design process as described on: "The main objective of this controller was to reduce drivetrain torsion." (pg. 2) "In order to implement a model-based controller to damp drive-train oscillations, a simplified turbine model including a drivetrain torsion DOF needs to be derived. Using a flexible drive-train model of the low-speed shaft (LSS), gear box with ratio Ng, and high speed shaft (HSS) as depicted in Figure 1, torque balances on either

side of the gear box are performed to develop the relationship between generated aerodynamic torque $\tau_a$, drive-train torsion and applied (control) generator torque $\tau_g$." (pg. 5) "To deal with the complexity of the LPV optimization process, the gridding (Wang and Seiler, 2018) technique is utilized, allowing existing conic solvers (Sturm, 1999) to find globally optimal solutions within a highly nonlinear domain and ill-conditioned problem (Ostergaard et al., 2009), in addition to creating finite dimensional LMI's from an originally infinite dimensional LPV model." (pg. 9) Additionally, an attempt to describe the consequence of this design methodology is given in Sections 3.3 and 3.4 "A sample time series plot of the controller performance is shown in Figures 8. Examining the plot, periods of drive-train damping are observed as compared with the baseline controller especially around t=125s, but there is not a clear overall difference for the entire time series. A statistical analysis of the variance for the load channels depicted in Figure (8) paints a clearer picture with LSS torsional moment (LSSMxa) of the DT controller having a 10.71% lower variance than the BL, the tower base side-to-side (TwrBsMxt) moment having a 30.54% lower variance, while the tower base fore-aft (TwrBsMyt) moment has a 19.95% larger variance than the BL controller. The reduction in the in-plane loads is due to the LPV controller applied generator torque aiming to maintain the rotor's angular velocity and minimize torsional oscillations between the LSS and HSS, all acting within the plane of the rotor." (pg. 19) "The LPV DT controller was able to reduce the LSS torsional DEL by 6.20%, the tower base side-to-side DEL by 14.77%, but with an unfortunate increase in tower base fore-aft DEL of 8.38%. The 5 increase in tower base fore-aft bending moment DEL is not surprising, as the main driver behind out-of-plane loads are driven by thrust force, requiring pitch actuation for substantial influence as has been shown in (Bossanyi and Hassan, 2000)." (pg. 24) The two-bladed, downwind... Now you suggest that you will perform your research on two-bladed turbines, but this does not become clear in the rest of the introduction. Additional clarification has been added in the form of "The two-bladed, downwind configuration that is the focus of an ongoing research grant (Loth et al., 2017) is thus far unexplored in the literature with regard to LPV control, but may provide additional

challenges and opportunities for this architecture. The contributions of this paper are twofold. First, an outline of a systematic tuning procedure utilizing a highly accurate aero-elastic turbine model (FAST) is presented for output feedback, drivetrain damping LPV torque control. The controller is synthesized using LMI constraints derived from an $H_\infty$ performance metric. Secondly, comparisons of controller derivation and performance to existing methods (Ostergaard et al., 2009), and implementation of the torque controller on a FASTv8 model of the aero-elastically scaled SUMR (Loth et al., 2017) is achieved. The control synthesis methodology outlined in this paper lends itself to a wind turbine plant regardless of turbine rating or number of blades. The controller can be obtained using aerodynamic basis functions (Mohammadpour and Scherer, 2012) and simplified linear models obtained from aero-elastic simulators to arrive at a working torque controller for a full degree of freedom (DOF), non-linear plant." (pg. 3) The contributions of this paper... Sentence is unreadable, rewrite. The sentence has been revised as presented in response 8i. The following paper -> this paper Suggestion has been incorporated. Are the aerodynamic basis functions + linearizations obtained and/or based on the full DOF non-linear plant? The aerodynamic basis functions are obtained on a limited DOF plant, which includes the drive-train torsion and generator DOF. Sentence has been revised as "Figure 2 shows these partial derivatives as functions of normalized free stream wind speed at selected turbine operating points along the turbine's operating trajectory in partial and full load conditions as generated by a plant with only the generator DOF enabled." (pg. 8) Modeling Introduce this section. What will you be doing in what section? In addition to an overall change in paper structure and organization that includes a re-wording of the subsection title to "2.2 LPV Drivetrain Model", an introduction has been added in the form of "Control design utilizes simplified plant models describing the dynamics of interest for feedback gain synthesis. In this section, a generalized three DOF plant model will be derived in the form of (1) using physics based principles and simple force balance calculations. The end goal is to obtain a below-rated torque controller with drivetrain damping and energy capture objectives which will be applied to the SUMR-D turbine. The SUMR is

a novel rotor design concept which aims to utilize morphing rotor technology in order to accomplish load alignment (Loth et al., 2017) such that structural design requirements are reduced. The SUMR-D design is a scaled version of the SUMR-13i design (Ananda et al., 2018), which will be used to validate the design process as a proof of concept field prototype. Specific details of the turbine model we use in the research follow in Section 3. While the application of the controller will be applied to the down-wind, two bladed SUMR-D rotor, the modeling and controller synthesis procedure is valid for any VSVP wind turbine model." (pg. 5) Which wind turbine (model) do you use in this paper? The wind turbine model has been introduced in Section 2.2 (pg. 5) as given in response 1i above. Furthermore, Section 3.1 gives further details of the SUMR-D rotor, see below for text. "Controller performance evaluation was conducted as applied to the Segmented Ultra-light Morphing Rotor – Demonstrator (SUMR-D). This rotor is a Gravo-Aeroelasticly Scaled (GAS) model (Loth et al., 2018; Kaminski et al., 2018) of the 100- meter blade SUMR-13i (Ananda et al., 2018; Martin et al., 2017; Martin and Zalkind, 2016; Zalkind et al., 2017) down-wind, two-bladed rotor with a pre-aligned rotor (Loth et al., 2017). This scaling method aims to fully capture the key dynamics of the full-scale model with an emphasis on matching the non-dimensional flapping frequency, moment ratios, flapping tip deflection and design tip speed ratio as represented by values given in Table 2." (pg. 18) Why do you use different symbols ($Q_a$ and $\tau_g$) for torques? Also $Q_{xx}$ and $\tau$ are not indicated in Figure 1. $Q_a$ has been changed to $\tau_a$ and is used to denote the aerodynamic torque generated by the wind exerted on the rotor, while $\tau_g$ is used to denote the electrical torque applied by the generator on the high speed shaft. The variables have been updated and included in Figure 1. Additionally, state-space system has been updated as seen in Figure 4 below.

Maybe introduce a new symbol for $\phi_r$- $\phi_g$ A new symbol of $\Delta\ddot{I}T$ denoting the difference between the azimuthal positions has been included. How do you obtain the aerodynamic torque gradients? From data of a real wind turbine? The aerodynamic torque gradients are obtained through FASTv8 linearization and an optimal tip speed ratio $\lambda_{opt}$ tracking trajectory. This has been made clearer through the explanation on (pg. 7) as

"Figure 2 shows these partial derivatives as functions of normalized free stream wind speed at selected turbine operating points along the turbine's operating trajectory in partial and full load conditions as generated by FASTv8 for a plant with only the generator DOF enabled." As well as, in the caption of Figure 2 in the form of "Values of partial derivatives for aerodynamic torque approximation (7) as generated through FASTv8 linearizations". Future work may look into obtaining basis functions from real wind turbine data. Definition of $Q_{HSS}$ is missing $Q_{HSS}$ has been added to the nomenclature section, and is defined within the Section 2.2 LPV Drivetrain Model (pgs.5-7). Directly substitute Eq. (6) in (5). This substitution has been incorporated into the paper (pg. 7) resulting in equation 8, given as seen in Figure 4 According to my derivation, the state-space system in (8) is incorrect. For example, according to Eq. (6), you miss a 1/Ng for the $C_d$ term in the (1,1) element. Or (6) is incorrect. A factor of 1/Ng has been added in the (1,1) element of (8), and additional checks have ensured correctness of equations. How do you obtain the partial derivatives? What is your approach? The partial derivatives are the basis functions/aerodynamic torque gradients, which are obtained through FASTv8 linearizations. The analytical expressions are obtained using first order Taylor Series expansions. This has been clarified in the text as "Figure 2 shows the numerical values of the partial derivatives representing the perturbations around the mean given in the first order Taylor Series expansion (7) as functions of normalized free stream wind speed at selected turbine operating points along the turbine's operating trajectory in partial and full load conditions as generated by FASTv8 for a plant with only the generator DOF enabled." (pg. 7) Why do you only consider 2 to 5 m/s wind? This is just the lower-region of the below-rated region, some wind turbines do not even operate at these wind speeds! 2-5 m/s wind speeds are considered because these wind speeds are defined as the below-rated operating range of the aerodynamically scaled rotor. The reasoning behind this wind speed range has been included in the paper as "This paper is focused on partial load operation (region 2) (Johnson et al., 2006) of the SUMR-D turbine, therefore, only the portion of the sensitivities falling between the cut-in 2 m/s and rated 5 m/s wind speeds are used in the construction of

the parameter varying functions corresponding to the novel rotor design and gravi-aero elastic scaled operating points, to be used in a research field test. These properties are scaled values of the SUMR-13i (Ananda et al., 2018) turbine, and applied to the structural, aerodynamic, and operational properties resulting in lower than normal operational set points. See Section 3.1 for more information." (pg. 8) Performance Vector Design "For this application ... measured scheduling parameter", you stated in you introduction that you would only use output feedback of the generator speed! Also the assumption of 0 error is not really feasible. This statement has been revised in Section 2.4 as "To achieve load reduction, rotor speed regulation is accompanied by an additional objective of reducing drivetrain torsion as compared to a baseline control architecture. The signals of which the control vector is composed, constitute the basis for the upper bound placed on the performance metric $\gamma\infty$ (16). The controller utilizes measured HSS angular velocity in addition to the measured free stream wind speed $V_\infty$ to schedule control gains throughout the envelope of turbine operation. This architecture provides increased performance with no additional sensors (as $V_\infty$ is assumed to be a readily available signal) making its application straightforward and its benefits more fruitful than an advanced control architecture requiring additional sensors not traditionally available on existing turbines." (pg. 9) The infeasibility of 0 error for measured inflow velocity is addressed within the paper. An additional description has been added in the form of "For the results presented in this paper, the hub height free stream wind speed $V_\infty$ is assumed to be perfectly measurable using a turbine-mounted anemometer, resulting in a measured bias error of 0 for the uncertainty $\delta_\theta$ in the measured scheduling parameter. The authors recognize that this assumption is not completely accurate, and in turn could prohibit the successful application of the control architecture to physical plants." (pg. 5) Cite Bossanyi for the drivetrain damper band pass-filter part. "The Design of Closed Loop Controllers" Citation has been added to the sentence "This shaping procedure is accomplished by multiplying the transfer function of the weighting functionWz1 (Bossanyi and Hassan, 2000) with the sensitivity function (13). The weighting function is centered at the drivetrain first eigenfrequency

$\Xi_{DT_{1P}}$, and shown in Figure 4. This provides the desired frequency response." (pg. 10) Remove Eq. 13. Equation 13 has been removed. Figure 4: I don't see a low-pass filter behavior in $z_2$. The description within Section 2.4 Performance Vector Design has been updated and clarified through the following additions "In addition to prioritizing drivetrain torsions, high frequency variations in the torque control signal are penalized using a first order weighting function $W_u$, shown in Figure 5, with a cutoff frequency set at 1.5 times above the 3P frequency to reduce control actuation for frequencies above that harmonic (16), as the 1P and 2P harmonics are the loads which contribute most to fatigue of a two bladed turbine necessitating control action within that bandwidth. The choice of cutoff frequency for this filter influences the final performance of the end controller. If the cutoff frequency is set too close to the 1P harmonic, the controller will not be able to regulate the rotor sufficiently, on the other hand, if the cutoff frequency is set too high the controller will control for high frequency excitations which could have the adverse effect of increasing fatigue loading on turbine components. In addition to determining the cutoff frequency, the DC gain of $W_u$ is also essential to final controller performance as this value governs the magnitude of the control signal as applied to the wind turbine plant." (pg. 11) The design of z ... 3 state system. What do you mean here? This statement is referring to the loop shaping process that is used to influence end controller performance. When frequency domain transfer functions are shaped using filters, the end result is a shaped transfer function with additional states from the original one. Additional information in the form of the designed transfer function has been provided and the description has been edited as "The design of the performance vector z is done using linearized plant models as obtained by FASTv8 and loop shaping procedures outlined in (Mohammadpour and Scherer, 2012; Bianchi et al., 2006) to obtain the desired characteristics for the performance vectors. The additional filters add states of their own to the final transfer functions, resulting in state-space systems consisting of state vectors not compatible with the original model order. Since the control synthesis process is performed in the time domain, the transfer functions (16), are converted to matrices $C_z$ and $D_{zu}$ to be used in the definition of the performance

vector z given in (1). The introduction of pole-zero pairs introduced during the shaping process result in a system with incompatible dimensions, as $x \in \mathbb{R}^{n \times 1} C \in \mathbb{R}^{p \times n}$. For this reason, reduction techniques (Laub et al., 1987; Laub, 1980; Gawronski and Juang, 1990; Moore, 1981) aim to eliminate states produced during the frequency domain multiplication having small singular values, in turn, having little influence on the system." (pg. 13) Comment on the results in Figure 6, what can be seen? Additional explanation has been added in the form of "The frequency responses depicted by the solid lines in Figure 6 show the final shaped systems $G_{z1}$ and $G_{z2}$ while the dashed lines depict the original transfer functions $G_p$ and $G_v$. Figure 6(a) shows the frequency response of the original transfer function from the inflow wind $V_\infty$ to the drive-train state (labeled as Original) as compared to the shaped system $G_{z1}$. The shaped system aims to decrease the gain from the wind to drivetrain at lower frequencies, and attempts to increase system gain near the 1P harmonic of the drivetrain as can be seen from the increase in spike magnitude near 35 rad/s. One thing that we need to keep in mind is that the drivetrain torsion is a rotating state, and we are attempting to control this state without a transformation to a stationary frame of reference. Given that the controller aims to control for 1P oscillations from a stationary frame of reference, the end result will be a reduction in higher order harmonic load reduction as it has been shown that the mapping of non-stationary to stationary loads, and visa-versa, results in harmonic distortion (van Solingen and vanWingerden, 2015). On the other hand, Figure 6(b) depicts a smoothing of system response across the bandwidth of interest with no influence on gain near the drivetrain eigenfrequency. One thing to point out is that during this process, the unity feedback sensitivity function S used does not account for a controller in the loop. Instead, the shaping aims to close the loop with unity gain feedback, increasing system gains near the eigenfrequency of the drivetrain, in addition to increasing the gain of the system for a bandwidth of inflow which will need to be controlled for during turbulent conditions. By shaping these systems to have ideal frequency responses without a controller in the loop, and in turn, using them as optimization metrics, the end result will be a controller which can be placed in the loop

achieving such responses as depicted in Figure 6." (pg. 12) See Figures 2 and 3 below

Complete paragraph "The loop shaping ... and step response", should be revised. I don't understand what's going on here. This has been further clarified as presented in response 5ii. (pg. 12)

Aborted Review: We carefully revised the remainder of the paper and hope the reviewer will re-consider the second version.

Gear Box ($N_g$)

HSS Side

LSS Side

$I_r, \Omega_r, \phi_r$

$\tau_a$

$\tau_g$

Generator

$I_g, \Omega_g, \phi_g$

**Fig. 1.** Figure 1

[Figure]

**Fig. 2.** Figure 6a

[Figure]

**Fig. 3.** Figure 6b

$$
\begin{bmatrix} \delta\dot{\Omega}_r \\ \Delta\dot{\phi} \\ \delta\dot{\Omega}_g \end{bmatrix} = \begin{bmatrix} \frac{\frac{\partial \tau_a}{\partial \Omega_r}\big|_{OP} - C_d/N_g}{I_r} & \frac{-k_d}{I_r} & \frac{C_d}{I_r N_g} \\ 1 & 0 & \frac{-1}{N_g} \\ \frac{C_d}{I_g N_g} & \frac{k_d}{I_g N_g} & \frac{-C_d}{I_g N_g^2} \end{bmatrix} \begin{bmatrix} \delta\Omega_r \\ \Delta\phi \\ \delta\Omega_g \end{bmatrix} + \begin{bmatrix} 0 & \frac{\partial \tau_a}{\partial \beta}\big|_{OP}\frac{1}{I_r} \\ 0 & 0 \\ \frac{-1}{I_g} & 0 \end{bmatrix} \begin{bmatrix} \delta\tau_g \\ \delta\beta \end{bmatrix} + \begin{bmatrix} \frac{\partial \tau_a}{\partial V_\infty}\big|_{OP}\frac{1}{I_r} \\ 0 \\ 0 \end{bmatrix} \delta V_\infty
$$

**Fig. 4.** State Space

---

## Author Comment (AC2) · 11 Aug 2018

Response to referee 2 comments

Thank you to the reviewers and editors for the comments and the opportunity to revise and improve our paper. We have made substantial revisions to the paper to address the comments from both reviewers. In this document, we explain how we addressed specific comments in our revision.

Abstract 1. "For such applications, Linear Parameter Varying (LPV) control provides a state-space approach to designing nonlinear controllers with robust performance" you

also synthesis an LPV controller. Why is this a nonlinear controller? i. The abstract has been fully revised to address this and other comments as "Wind turbine fatigue damage can be greatly reduced through the application of Linear Parameter Varying (LPV) control architectures as compared to traditional control methodologies. LPV control theory facilitates optimal controller synthesis considering various objectives; in this work, we apply LPV control theory to a MIMO LPV model of a nonlinear turbine plant using Linear Matrix Inequalities (LMIs) and convex optimization solvers to produce the LPV controller. The application of the torque controller to a down-wind, two bladed, scaled, Segmented Ultra-light Morphing Rotor (SUMR) results in improved turbine performance and reduced damage equivalent load (DEL) accumulation during turbulent inflow. The reduction in DEL is attributed to the limit placed on the exogenous disturbances' effect on performance channels stemming from $H_\infty$ LMI formulation." ii. The controller is nonlinear because of the relationship between rotor angular velocity $\Omega_r$ and aerodynamic torque $\tau_a$. Details describing this relationship are presented in (Johnson et al., 2006, Martin et al., 2017). 2. "LPV uses multi-input multi-output (MIMO) model with a.." please check grammar (e.g Linear Parameter Varying uses . . . what does it mean?) i. We have addressed the comment. Please see revised abstract given in Response1i. Introduction 1. "robustness of H control" what do you mean? i. The paragraph has been revised as "LPV control theory provides guarantees of stability despite the presence of plant parameter uncertainty and/or uncertainties in the measured scheduling parameter (Shamma, 1988; Zhao and Nagamune, 2017; Sato and Peaucelle, 2013; Sato et al., 2010). It utilizes robust control theory (Levine, 2011) during the construction of the LMI constraints for the optimization problem, providing guarantees of stability in the presence of deviations from model parameters used during control synthesis, and similar guarantees of stability in the presence of exogenous disturbances the plant may be exposed to during operation. Conservative performance can be an issue, given significant deviations from the design operating point which can be addressed through LPV control theory's potential to create less conservative controller response by appropriate scheduling of the plant's dynamics. However, given the

degree of nonlinearity for variable speed, variable pitch (VSVP) wind turbines, solving the optimization problem is non-trivial given a system with a large number of states to be considered during the MIMO controller synthesis. To deal with the complexity of the LPV optimization process, the gridding (Wang and Seiler, 2018) technique is utilized, allowing existing conic solvers (Sturm, 1999) to find globally optimal solutions within a highly nonlinear domain and ill-conditioned problem (Ostergaard et al., 2009), in addition to creating finite dimensional LMI's from an originally infinite dimensional LPV model." (pg. 2) 2. P2, line 21, "a LPV" should be "an LPV" i. Edit has been addressed in this section as well as throughout the paper. Controller Derivation 1. P6, quality of the figure should be improved Quality of all figures has been addressed. Specifically, Figure 1 on pg. 6 has been updated and is shown below in Figure 1

2. "only the portion of the sensitivities falling between the cut-in (2 m/s) and rated (5 m/s) wind speeds will be used in the construction of the parameter varying functions" a rated speed of 5 m/s sounds strange. Is that for the novel rotor? If yes, this rotor model should be introduced. i. In order to reduce the confusion these low wind speeds may cause the reader, they have been normalized as presented in Table 1 (pg. 7) and Figure 2 (pg. 8). Additionally, further clarification has been added to reduce confusion about the turbine model used in this paper in the form of "Control design utilizes simplified plant models describing the dynamics of interest for feedback gain synthesis. In this section, a generalized three DOF plant model will be derived in the form of (1) using physics based principles and simple force balance calculations. The end goal is to obtain a below-rated torque controller with drivetrain damping and energy capture objectives which will be applied to the SUMR-D turbine. The SUMR is a novel rotor design concept which aims to utilize morphing rotor technology in order to accomplish load alignment (Loth et al., 2017) such that structural design requirements are reduced. The SUMR-D design is a scaled version of the SUMR-13i design (Ananda et al., 2018), which will be used to validate the design process as a proof of concept field prototype. Specific details of the turbine model we use in the research follow in Section 3. While the application of the controller will be applied to the down-wind, two

bladed SUMR-D rotor, the modeling and controller synthesis procedure is valid for any VSVP wind turbine model." (pg. 5) ii. "This paper is focused on partial load operation (region 2) (Johnson et al., 2006) of the SUMR-D turbine, therefore, only the portion of the sensitivities falling between the cut-in 2 m/s and rated 5 m/s wind speeds are used in the construction of the parameter varying functions corresponding to the novel rotor design and gravi-aero elastic scaled operating points, to be used in a research field test. These properties are scaled values of the SUMR-13i (Ananda et al., 2018) turbine, and applied to the structural, aerodynamic, and operational properties resulting in lower than normal operational set points. See Section 3.1 for more information." (pg. 8) 1. P13, figure 7 not clear i. All Figure quality has been addressed. ii. Updated Figure shown in Figure 2 below

2. For performance channels it doesn't make sense to present the phase i. Performance system phase has been removed from Figure 6. Performance Vector Design 1. Eq 15 with the previous S, really doesn't make sense. I stopped reading the paper after this point. While the ACC paper from the same authors was easy to read, correct and well organized this paper seems to be the opposite. i. Additional clarification for the sensitivity function has been added in the form "This shaping procedure is accomplished by multiplying the transfer function of the weighting function $W_{z1}$ (Bossanyi and Hassan, 2000) with the sensitivity function (13). The weighting function is centered at the drivetrain first eigenfrequency $\Xi_{DT_{1P}}$ , and shown in Figure 4. This provides the desired frequency response." (pg. 10)

Aborted Review We have thoroughly revised the paper and hope this reviewer and other readers find the revised version easier to read, correct, and better organized.
* * *
Gear Box ($N_g$)

HSS Side

LSS Side

$I_r, \Omega_r, \phi_r$

$\tau_a$

$\tau_g$

Generator

$I_g, \Omega_g, \phi_g$

**Fig. 1.** Drive Train Model

[Figure]

**Fig. 2.** Time Series